# Whole-body analysis of TRPML3 (MCOLN3) expression using a GFP-reporter mouse model reveals widespread expression in secretory cells and endocrine glands

Barbara Spix[1☯], Andrew J. Castiglioni[2☯], Natalie N. Remis[2,3☯], Emma N. Flores[2,4], Philipp Wartenberg[5], Amanda Wyatt[5], Ulrich Boehm[5], Thomas Gudermann[1], Martin Biel[6], Jaime García-Añoveros[2,3,4,7]*, Christian Grimm[1]*

1 Walther Straub Institute of Pharmacology and Toxicology, Faculty of Medicine, Ludwig-Maximilians-University, Munich, Germany, 2 Department of Anesthesiology, Northwestern University Feinberg School of Medicine, Chicago, Illinois, United States of America, 3 Integrated Graduate Program in the Life Sciences (IGP), Northwestern University Feinberg School of Medicine, Chicago, Illinois, United States of America, 4 Northwestern University Interdepartmental Neuroscience (NUIN) graduate program, Chicago, Illinois, United States of America, 5 Center for Molecular Signaling (PZMS), Experimental Pharmacology, Saarland University, Homburg, Germany, 6 Department of Pharmacy, Ludwig-Maximilians-University, Munich, Germany, 7 Departments of Neurology and Neuroscience, and Hugh Knowles Center for Clinical and Basic Science in Hearing and Its Disorders, Northwestern University Feinberg School of Medicine, Chicago, Illinois, United States of America

☯ These authors contributed equally to this work.
* anoveros@northwestern.edu (JGA); christian.grimm@med.uni-muenchen.de (CG)

**Data Availability Statement:** All relevant data are within the paper and its Supporting Information files.

## Abstract

TRPML3 (mucolipin 3, MCOLN3) is an endolysosomal cation channel belonging to the TRPML subfamily of transient receptor potential channels. Gain-of-function mutations in the *Trpml3* gene cause deafness, circling behavior and coat color dilution in mice due to cell death of TRPML3-expressing hair cells of the inner ear or skin melanocytes, respectively. Furthermore, TRPML3 was found to play a role in the long term survival of cochlear hair cells (its absence contributing to presbycusis), in specialized giant lysosomes that neonatal (birth to weaning) enterocytes used for the uptake and digestion of maternal milk nutrients, and in the expulsion of exosome-encased bacteria such as uropathogenic E. coli, infecting bladder epithelial cells. Recently, TRPML3 was found to be expressed at high levels in alveolar macrophages and loss of TRPML3 results in a lung emphysema phenotype, confirmed in two independently engineered *Trpml3* knockout lines. TRPML3 is not ubiquitously expressed like its relative TRPML1 and thus cellular expression of TRPML3 on a whole-tissue level remains, with the exceptions mentioned above, largely elusive. To overcome this problem, we generated a τGFP reporter mouse model for TRPML3 and compared expression data obtained from this model by immunofluorescence on tissue sections with immunohistochemistry using TRPML3 antibodies and in situ hybridization. We thus uncovered expression in several organs and distinct cell types. We confirmed TRPML3 expression in both neonatal and adult alveolar macrophages, in melanocytes of hair follicles and glabrous skin, in principle cells of the collecting duct of the neonatal and adult kidney, and in olfactory sensory neurons of the olfactory epithelium, including its fibres protruding to the glomeruli of

**Funding:** Financial Support: RO1 DK111032 and R01 DC015903 (to JGA), T32 NRSA NS041234 (to AJC and ENF), F31 NRSA DC010529 (to NNR), German Research Foundation (GRK2338 P08 to CG and MB, P09 to TG, SFB/TRR152 Z02 to UB, P04 to CG, P12 to MB, P15 and the German Center of Lung Research, DZL, to TG). The funders had no role in study design, data collection and analysis, decision to publish, or preparation of the manuscript.

**Competing interests:** The authors have declared that no competing interests exist.

the olfactory bulb. Additionally, we localized TRPML3 in several glands including parathyroid, thyroid, salivary, adrenal, and pituitary gland, testes and ovaries, suggestive of potential roles for the channel in secretion or uptake of different hormones.

## Introduction

Transient receptor potential (TRP) cation channels are fundamental for many biological processes throughout the body. Dysfunctions or loss of these channels are linked to several diseases in mice or humans, e.g. focal segmental glomerulosclerosis [1], olmsted syndrome [2], scapuloperoneal hereditary motor neuropathy [3], transient neonatal hyperparathyroidism (TNHP) [4], congenital stationary night blindness [5], western pacific amytrophic lateral sclerosis (ALS) and parkinsonism dementia (PD) [6], intellectual disability and epilepsy [7], progressive familial heart block type I (PFHBI) [8], hypomagnesia with secondary hypocalcemia [9], familial episodic pain syndrome [10], autosomal dominant polycystic kidney disease [11], lung emphysema [12, 13], mucolipidosis type IV [14], congenital deafness and vestibular dysfunction [15], age-related hearing loss (presbycusis) [16], intestinal pathology and failure to thrive in neonates [17], and many more [18, 19]. Several TRP channels are currently being explored by academia and pharmaceutical industry as potential drug targets for different indications [20–22]. Nevertheless the physiology and pathophysiology of a number of TRP channels remains largely unexplored. In this study, we performed a tissue expression analysis of TRPML3, which belongs to the MCOLN (TRPML) subfamily, one of the six subfamilies within the TRP channel superfamily. TRPML channels are different from most other TRP channels as they are mainly expressed intracellularly, in organelles of the endolysosomal system such as early endosomes, recycling endosomes, late endosomes and lysosomes, and TRPMLs have been shown to be involved in endocytosis, fusion/fission processes, trafficking, secretion, lysosomal exocytosis and autophagy [23–26]. Currently available tissue expression studies are mainly based on RT-qPCR [27, 28]. Using this approach, Cuajungco et al. [27] found relatively high TRPML3 expression in the brain, thymus, lung, kidney and spleen, whereas it was quite low in cerebellum, eye, heart, liver, pancreas, stomach, colon and testis. Also by RT-qPCR, Samie et al. [28] claimed a rather high expression of TRPML3 in thymus, lung, kidney, colon and spleen compared to the other analyzed organs. Using in situ hybridizaton, immunohistochemistry and RT-qPCR, Nagata et al. [29] and Castiglioni et al. [30] found TRPML3 in stria vascularis, hair cells, vomeronasal and olfactory receptor neurons, while being largely absent in other sensory neurons such as somatosensory neurons, retinal neurons and taste receptor cells [30]. Remis et al. [17] detected TRPML3 in the characteristic enterocytes of neonatal (birth to weaning) small intestine, while it was not detectable in adult (post weaning) enterocytes. Furthermore, high TRPML3 expression was reported for melanocytes. Melanocytes are also lost in mice with the varitint-waddler (Va) phenotype (coat color dilution) due to a mutation in the mouse *Trpml3* gene (A419P) [31].

In our study, we used in situ hybridization (ISH) with two nonoverlapping cRNA antisense probes (5' *Trpml3* and 3' *Trpml3*), immunohistochemistry (IHC) with two antisera raised against different regions of TRPML3 (NT and CT1), as well as tissue sections from a *Trpml3*^IRES-Cre/eR26-τGFP reporter mouse model [32] to evaluate on the one hand hitherto reported TRPML3 expression in several organs and to discover new expression locations on a cellular level in different organs and tissues. We analyzed adult mouse organs (P48), but also organs and tissues from prenatal (E18) and neonatal (P2, P7, P13) mice to better understand developmental dynamics of TRPML3 in selected organs.

## Materials and methods

### Animals

All animal handling was in strict accordance with the *Guide for the Care and Use of Laboratory Animals* published by the National Institutes of Health and were approved by the Institutional Animal Care and Use Committee at Northwestern Univerisity and Saarland University. Mice were housed in the barrier rooms of Northwestern University's or Saarland University's animal facilities. We obtained tissues from either *Trpml3*$^{\text{IRES-Cre/eR26-τGFP}}$ reporter mice or CD1 mice (Charles River), or from *Trpml3*$^{-/-}$ and *Trpml3*$^{+/+}$ littermates with a genetic background of ~75% C57BL/6 and ~25% Sv129/Ola. See Castiglioni et al. [30] for detailed generation and genotyping information of *Trpml3*$^{-/-}$ mice. Generation of *Trpml3*$^{\text{IRES-Cre/eR26-τGFP}}$ reporter mice was described previously [32–34]. Briefly, two mouse lines were used to generate the *Trpml3*$^{\text{IRES-Cre/eR26-τGFP}}$ reporter mice: *Trpml3*-IRES-Cre mice were crossed to Cre-dependent ROSA26-CAGS-τGFP (eR26-τGFP) fluorescent reporter mice. For all animal experiments, mice were deeply narcotized using a mixture of ketamine/xylazine. The depth of anesthesia was confirmed by the absence of any reflexes before moving on with the experimental procedures.

### Antisera characterization

Our present study uses triple controlled immunohistochemistry and (double) immunofluorescence to determine the tissue and subcellular expression pattern of TRPML3 protein. We employ antisera raised against distinct regions of TRPML3. We also compare TRPML3 immunoreactivities to available in situ hybridization (ISH) analyses. Finally, we determine which immunoreactivities are absent from the tissues of a *Trpml3*$^{-/-}$ mouse. See Castiglioni et al. [30] for detailed TRPML3 antibody information and complete prior characterization. Antibodies used in this study include: TRPML3-NT (rabbit polyclonal, Sigma Cat. M7570, Lot 067K4822); TRPML3-CT1 (rabbit polyclonal, kind gift of David Clapham and Markus Delling, [31]); Tyrosinase M19 (goat polyclonal, Santa Cruz Cat. sc-7834, Lot L1208); F4/80 (rat monoclonal, AbD serotec Cat. MCA497GA, clone A3-1); Aquaporin 1 (rabbit polyclonal, Alpha Diagnostics Cat. AQP11-A, Lot 117980A12.2); Aquaporin 2 (goat polyclonal, kind gift of Daniel Batlle); Chromogranin A (rabbit polyclonal, abcam Cat. 15160); GFP (chicken, Thermo Fisher Scientific A10262); CD8 alpha (rabbit monoclonal, abcam Cat. 217344, clone EPR21769); CD45R (rat monoclonal, BD Biosciences Cat. 550286, clone RA3-6B2). For immunofluorescence the following secondary antibodies were used: Donkey anti-chicken-488 IgG (Jackson Immunoresearch Cat. 703-225-155); Donkey anti-rat-Cy3 (Jackson Immunoresearch Cat. 712-165-153); Donkey anti-rabbit-Cy5 Jackson Immunoresearch Cat. 711-175-152).

### Tissue processing

We used unfixed tissues for *in situ* hybridization and fixed tissues for immunohistochemistry. Unfixed tissues were dissected, embedded in OCT and immediately snap frozen in dry ice isopentane. For fixed adult tissues, we transcardially perfused the animal with 2% paraformaldehyde (PFA), dissected out the organs, postfixed for 1 hour in 2% PFA, and rinsed 3 times in 1X PBS. We then took the tissue through a sucrose gradient (1 hour each 5%, 10%, 20%), ending with an overnight incubation in 20% sucrose and 50% OCT (Tissue-Tek, Sakura). We mounted the tissue in OCT (Tissue-Tek, Sakura) and froze it on dry ice. PFA was always prepared fresh from powder stocks directly prior to use. Consistency in fixation was very important as we often saw background autofluorescence that we could ascribe to over or inconsistent fixation.

For tissues from *Trpml3*<sup>IRES-Cre/eR26-τGFP</sup> reporter mouse we transcardially perfused the animal with PBS, followed by 4% PFA, and removed the organs for a postfixation of 3 hours in 4% PFA. Tissues were incubated in 18% sucrose solution overnight and frozen in OCT (Tissue-Tek, Sakura). Finally, we prepared 10μm cryosections of all organs. For the brain and pituitary gland 14μm cryosections were prepared. For all figures, tissues were obtained from two adult mice, one female and one male.

## Immunohistochemistry

With noted exceptions, we performed all TRPML3 immunohistochemistry using the tyramide signal amplification system (TSA, Alexa488 tyramide, Invitrogen). Where noted, we used the ABC/DAB (avidin-biotin complex with diaminobenzidine reaction) signal amplification system (Vector). Otherwise, primary Abs were detected by using conventional Texas Red (Jackson Immuno) or Alexa568 (Invitrogen) labeled secondary antibodies. Following are brief protocols of our immunohistochemistry techniques.

**Tyramide signal amplification.** Fixed sections: Air dry for 15 min, vacuum dry for 15 additional min, and rinse 3x 5 min in 1X PBS (Lonza). Unfixed sections: Thaw briefly, postfix for 10 minutes in freshly prepared 2% paraformaldehyde, and then rinse 3x 5 min in 1X PBS. Retrieve antigens by incubating in 10mM sodium citrate, pH 6 with 0.25% Triton for 20 min at 92˚C. After cooling for 30 min at room temperature, rinse 3x 5 min in 1X PBS. Quench endogenous peroxidase by incubating in 1% $H_2O_2$, 10% methanol, in 1X PBS for 30 min. Rinse 3x 5 min in 1X PBS. Block for 1 hour in 1% TSA block solution (Invitrogen). Incubate with primary antibody (NT, 1:5000–1:10,000; CT1 1:1000–1:2000; in 1% TSA block solution (no azide)) overnight at 4˚C. The next day, rinse 4x 10 min in 1X PBS. Incubate with secondary antibody (1:100, goat anti-rabbit, Invitrogen) in 1% TSA block solution for 1 hr protected from light. Rinse 4x 10 min in 1X PBS. Proceed with tyramide labeling reaction in groups of 5–6 slides. Apply tyramide (1:150, Invitrogen) working solution to slide. Incubate for 10 min at room temperature protected from light. Rinse 3 x 10 min in 1X PBS, in dark. Add DAPI (1μM) for 10 min. Rinse 2x 10 min in 1X PBS. Rinse with ddH$_2$O. Mount with Prolong Gold (Invitrogen).

For double immunohistochemistry using TSA to detect one of the TRPML3 antibodies, we obtained our best results when performing TSA detection of the TRPML3 antibody, followed by the conventional detection of the second primary antibody.

**ABC+DAB signal amplification.** Air dry fixed sections for 15 min and vacuum dry for 15 additional min. Rinse 3x 5 min in 1X PBS (Lonza). Thaw unfixed sections briefly, then postfix for 10 minutes in freshly prepared 2% paraformaldehyde. Rinse 3x 5 min in 1X PBS. Retrieve antigens by incubating in 10mM sodium citrate, pH 6 with 0.25% Triton for 20 min at 92˚C. After cooling for 30 min at room temperature, rinse 3x 5 min in 1X PBS. Quench endogenous peroxidase by incubating in 1% $H_2O_2$, 10% methanol, in 1X PBS for 30 min. Rinse 3x 5 min in 1X PBS. Block for 2 hours in 10% normal goat serum, in 1X PBS. Incubate primary antibody (NT, 1:2000; CT1, 1:1000) in 10% normal goat serum block + 0.1% triton (no azide) overnight at 4˚C. The next day, rinse 4x 10 min in 1X PBS + 0.1% triton. Incubate with biotinylated secondary antibody (1:200, goat anti-rabbit, Vector) in 10% normal goat serum block + 0.1% triton for 1 hr. Rinse 4x 10 min in 1X PBS + 0.1% Triton, 50 rpm. Prepare ABC solution (Vector) 30 min prior to use. Incubate in ABC solution for 1 hr. Rinse 3x 5 min in 1X PBS. Incubate with DAB solution (Sigma) for at least 7 min until tissue turns light brown. Rinse 3x 5 min in 1X PBS. Add DAPI (1μM) for 10 min. Rinse 2x 10 min in 1X PBS. Rinse with ddH$_2$O. Mount using Prolong Gold (Invitrogen).

For double immunohistochemistry using ABC+DAB to detect one of the TRPML3 antibodies, we obtained our best results when performing ABC+DAB detection of the TRPML3 antibody, followed by the conventional detection of the second primary antibody.

## Immunofluorescence

Tissue cryosections from $Trpml3^{\text{IRES-Cre/eR26-τGFP}}$ reporter mice were air dried for 15 min and then washed 3x 5 min in 1X PBS. After 1 h blocking in a solution of 1X PBS containing 10% normal donkey serum (Jackson Immunoresearch Cat. 017-000-121), 3% BSA and 0,3% Triton-X-100, tissue sections were incubated with the primary antibodies overnight at 4˚C. The following day, we rinsed the sections 3x 5 min in 1X PBS containing 0,05% Tween-20 (PBST) and incubated them with the secondary antibodies, diluted 1:500, for 2 h at room temperature. Nuclei were stained for 5 min using 2ug/ml Bisbenzimide solution (Sigma, B1155) before 3x 5 min final washes in PBST and mounting in Fluoromount-G (Biozol, SBA-0100-01).

## In situ hybridization

We performed in situ hybridization on cryostat sections of snap frozen, unfixed tissues from CD1, $Trpml3^{+/+}$, and $Trpml3^{-/-}$ mice using protocols previously described [29, 35, 36]. Freshly dissected and unfixed tissues were immediately snap frozen by dipping in isopentane cooled to −30˚C with dry ice and sectioned (10–12 μm). For ISH, we used two non-overlapping cRNA probes for mouse $Trpml3$ mRNA (Genbank ID NM_134160). These are a 5' probe, which corresponds to nucleotides 179–723 (from codon 60 at the end of exon 1 to codon 240 at the end of exon 5) and a 3' probe, which corresponds to nucleotides 1005–1594 (from codon 335 in the middle of exon 8 to codon 531 in the middle of exon 12, the last exon).

We PCR amplified these cDNA fragments from mouse inner ear or CVP mRNA and TA-cloned them into vector pCRII. We generated digoxigenin-labeled antisense and sense (control) cRNA probes using the DIG-RNA labeling kit (Roche) according to the manufacturer's instructions. Sections were hybridized with antisense or sense probes as previously described [36]. Sections were mounted for observation. Only cell types that labeled with both the 5' and 3' $Trpml3$ probes were considered positive for $Trpml3$ mRNA.

## Image acquisition and analysis

We acquired images using either a Nikon E600 pan fluorescence microscope (20x 0.75 N.A., 60x 1.4 N.A., or 100X 1.4 N.A. objectives) equipped with a CCD camera (SPOT RC-Slider) or a Zeiss LSM 510 confocal microscope (63x 1.4 N.A. or 100x 1.46 N.A. objectives) or a Leica SP5 confocal microscope (63x, 1.4 N.A. objective). When comparing wild type and knockout immunoreactivities, we captured images under identical conditions. In practice, this meant capturing images with identical exposure settings (pan fluorescence) or identical laser and gain settings (confocal). For even illumination, we flat field corrected and white balanced the color (SPOT RC-Slider) camera prior to acquiring DIC images. Post acquisition, we identically processed image pairs of wild type tissues and their corresponding knockout controls. This included deconvolution of pan fluorescence images as well as adjustment for brightness and contrast of all images. Confocal images shown in this paper represent a maximum intensity projection of either two or three Z-stack image slices. We used ImageJ for all post acquisition processing, with the exception of deconvolution, for which we used MetaMorph.

Images showing tissues isolated from $Trpml3^{\text{IRES-Cre/eR26-τGFP}}$ reporter mice were aquired using a Zeiss AxioScan.Z1 slide scanner and processed using the ZenBlue software.

## RT-qPCR

For RT-qPCR (reverse transcription of RNA followed by quantitative polymerase chain reaction), we used tissues from instantly killed animals which were not transcardially perfused. All tissues were dissected as quickly and as cleanly as possible and immediately snap frozen on dry

ice, until homogenized in Trizol (Invitrogen). We homogenized all tissues in Trizol using a tissue homogenizer, and performed RNA isolation according to the manufacturer's instructions. RNA concentration was determined by UV absorption ($OD_{260}$). This value helped determine the volume of RNA used per RT reaction, with the goal to reverse transcribe 1 μg of total RNA per reaction. Prior to reverse transcription, we subjected the 1 μg of total RNA to DNaseI treatment (Invitrogen) to eliminate genomic DNA according to the manufacturer's protocol. This DNaseI-treated 1 μg of total RNA was then subjected to first strand cDNA synthesis using Superscript III reverse transcriptase (Invitrogen) according to the manufacturer's protocol.

We performed RT-qPCR using a Mastercycler Realplex2 machine (Eppendorf) on ~100 ng (2 μl of a 26 μl RT reaction) of first strand cDNA using SYBR Green PCR Mastermix (Applied Biosystems) in triplicate, according to the manufacturer's instructions. The following primers (IDT) were designed on mouse sequence and used in qPCR: Trpml3ex8f, 5' ATGGAGTTCAT CAACGGGTG; Trpml3ex9r, 5' ATAGTTGACGTCCCGAGAAG; 18Sf, 5' TTGACGGAAGGGC ACCACCAG; 18Sr,5' GCACCACCACCCACGGAATCG. Melting curve analysis and gel electrophoresis of PCR products indicated single products of the correct size for each primer pair used. The $Trpml3^{-/-}$ mouse does not contain the binding site for primer ex8f and prior qPCR analysis [30] on $Trpml3^{-/-}$ mice using primers ex8f and ex9r did not detect any product from $Trpml3^{-/-}$ tissue.

We determined the amount of $Trpml3$ mRNA, relative to VNO, in different tissues using the $2^{-(\Delta\Delta CT)}$ method where $\Delta\Delta CT$ represents: $(CT_{Trpml3tissue}-CT_{18Stissue})-(CT_{Trpml3VNO}-CT_{18SVNO})$. We report all $Trpml3$ mRNA amounts relative to VNO because VNO RNA had the lowest $18S$ CT value indicating the highest amount (i.e., least degradation) of RNA. All of our present analysis indicates that VNO has the highest density of TRPML3 mRNA and protein per integral organ.

## Results

### Distribution of *Trpml3* mRNA in major adult organs

Previously, we reported the expression patterns and developmental dynamics of TRPML3 in inner ear, vomeronasal, and olfactory sensory organs [29, 30], as well as in neonatal intestinal enterocytes [17]. In this study, we analyzed further mouse tissues by RT-qPCR and detected *Trpml3* mRNA in adult retinal pigmented epithelium (RPE), thymus, back skin, kidney, lung, and, at very low levels, spleen and liver (Fig 1A). Previously, we were unable to detect *Trpml3* in the neuronal retina when separated from the RPE [30]. In our present analysis, we could not detect *Trpml3* mRNA in adult cerebellum, heart, muscle, stomach, small intestine and colon (Fig 1A). Our results are consistent with previous RT-PCR and RT-qPCR analyses that detected *Trpml3* mRNA expressed in RPE cell lines [37], and in the adult (8–10 weeks) eye, thymus, spleen, liver, kidney, and lung [28]. To study the TRPML3 distribution and developmental dynamics in more detail we now examined lung, skin and kidney by in situ hybridization (ISH) with two nonoverlapping cRNA antisense probes (5' *Trpml3* and 3' *Trpml3*; locations shown in Fig 1B) and by immunohistochemistry (IHC) with two antisera (NT and CT1; locations shown in Fig 1C) raised against different regions of TRPML3 using tissues from both $Trpml3^{+/+}$ and $Trpml3^{-/-}$ mice to control for nonspecific immunoreactivities. In addition, we used $Trpml3^{IRES-Cre/eR26-\tau GFP}$ reporter mice to visualize τGFP expression and correlate it with the presence of TRPML3 in several tissues and cells (fusion of GFP with the microtubule-associaited protein Tau enables labeling of axons). These analyses include lung, olfactory bulb, olfactory sensory organs, skin, kidney, thyroid/parathyroid, salivary gland, adrenal gland, pituitary gland, ovary and testes, and thymus.

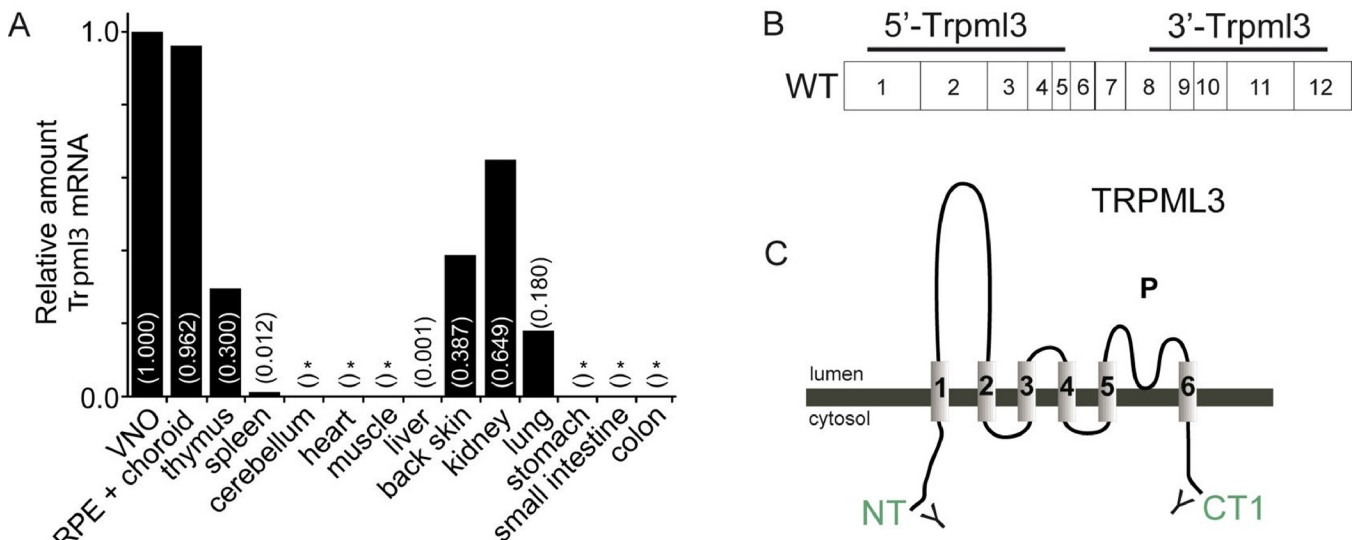

**Fig 1. Distribution of *Trpml3* mRNA in major adult organs. (A)** *Trpml3* levels detected by RT-qPCR using primers in exon 8 and exon 9 of *Trpml3*, normalized to *18S* rRNA, and displayed (in parentheses) relative to normalized *Trpml3* levels in vomeronasal organ (VNO). RT-qPCR in cerebellum, heart, muscle, stomach, small intestine and colon did not detect *Trpml3* reliably in three qPCR trials (*, indicates *Trpml3* not detected in 0/3, 1/3 or 2/3 qPCR trials). All tissue is from an adult CD1 mouse, with the exception of VNO, RPE + choroid, cerebellum, and muscle. Back skin tissue is from a P27 CD1 mouse because hair follicle growth (anagen II) is synchronized at this age. **(B)** Illustration of two non-overlapping cRNA antisense probes (5'*Trpml3* and 3'*Trpml3* locations), that were used for in situ hybridization (ISH). **(C)** NT and CT1 (green) are the two antisera used to detect TRPML3 in different tissues by immunohidtochemistry.

## Alveolar macrophages of neonatal and adult express TRPML3

Spix et al. [32] has previously described the expression pattern of TRPML3 in the lung using the *Trpml3*[IRES-Cre/eR26-τGFP] reporter mice by FACS analysis and immunofluorescence of tissue sections. We found τGFP positive (= TRPML3+) cells co-expressing the macrophage marker F4/80 in lung tissue (Fig 2A) as well as in bronchoalveolar lavage, mainly containing alveolar macrophages (Fig 2B).

Because lungs begin to function at birth we wondered if *Trpml3* expression correlates with the onset of lung function. We therefore examined *Trpml3* mRNA levels in lung tissue of various stages from prenatal (E18) to adult by RT-qPCR and ISH. Our RT-qPCR analysis indicated that *Trpml3* mRNA levels in the lung are approximately 10-fold less abundant in embryonic (E18) lung compared to adult lung (Fig 2C). Thus, the increase in *Trpml3* mRNA following birth correlates with the onset of lung function. When we analyzed sections of neonatal (P2) lung with both nonoverlapping antisense ISH probes to *Trpml3* mRNA, we found signals in cells scattered throughout the lung (Fig 2D and 2E). This signal was not present in lung sections analyzed with control sense ISH probes or lung sections from the *Trpml3*[-/-] mouse analyzed with antisense ISH probes to *Trpml3* mRNA.

When we performed immunohistochemistry on sections of adult (P48) lungs, we saw immunoreactivity with both NT and CT1 antisera. This immunoreactivity was present on cells also scattered throughout the *Trpml3*[+/+] lungs (Fig 2F and 2G) but not *Trpml3*[-/-] lungs (Fig 2J and 2K). Upon closer examination of these immunoreactive cells, we observed that they appeared distinct and often semi-detached from the cells that constitute the alveolar sac (Fig 2H, 2I and 2L). The developmental shift in mRNA expression in whole lung, combined with the location and morphology of TRPML3-immunoreactive cells within the lung suggested that TRPML3-expressing cells were alveolar macrophages. When we performed co-immunohistochemistry on sections of adult (P48) lung using NT antisera and a monoclonal antibody

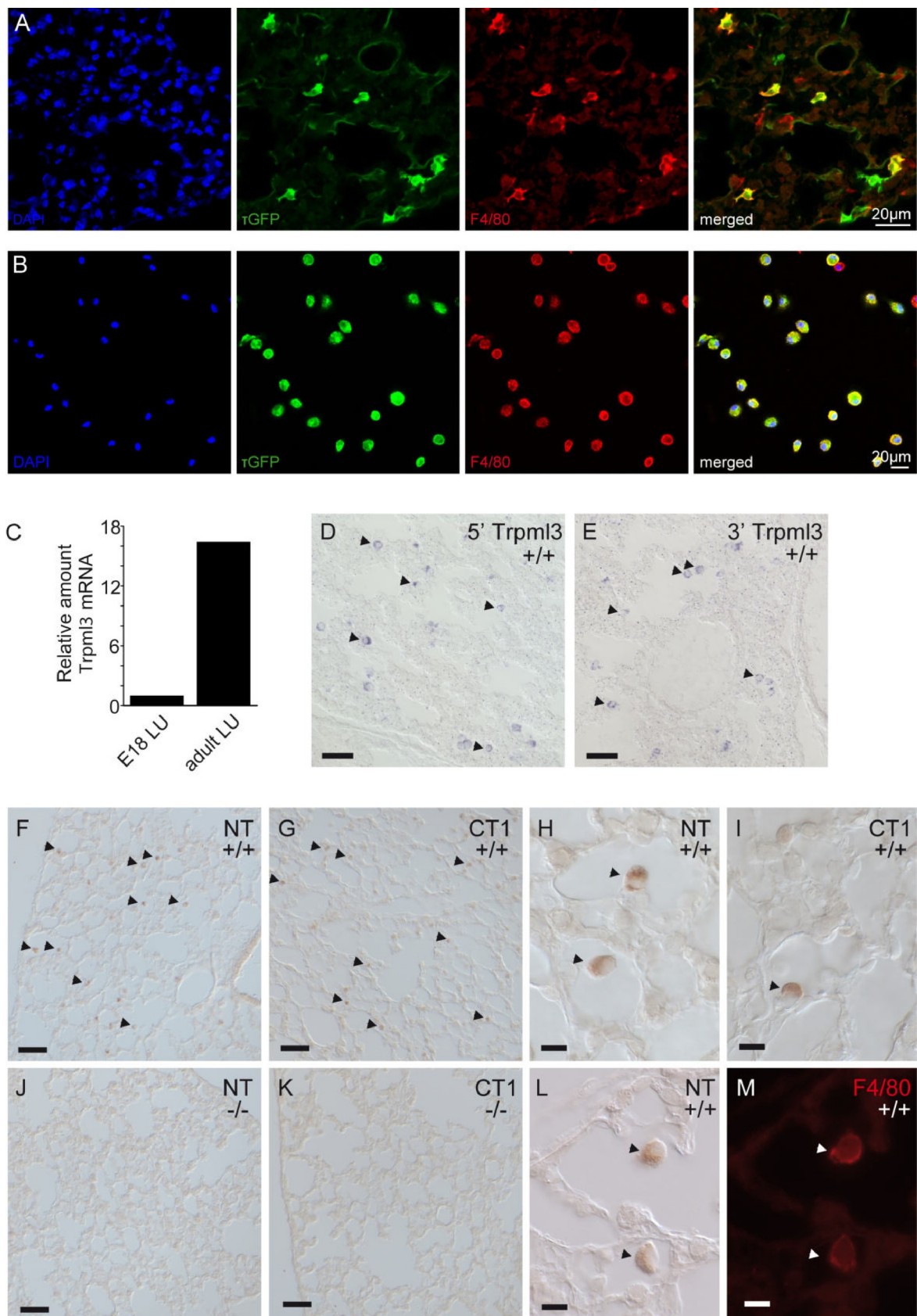

**Fig 2. Alveolar macrophages of neonatal and adult express TRPML3.** Immunofluorescence images using F4/80 antibody as macrophage marker **(A)** in 10 μM lung cryosections or **(B)** on cells isolated by bronchoalveolar lavage. Images clearly show co-localization of τGFP and F4/80 confirming TRPML3 expression in lung macrophages. **(C)** *Trpml3* levels in lung increase from birth to adult. *Trpml3* detected by RT-qPCR and normalized as in Fig 1. Levels are displayed relative to normalized *Trpml3* in E18 lung. **(D, E)** In situ hybridization on sections of neonatal (P2) lungs from *Trpml3*$^{+/+}$ mice shows a pattern of scattered positive cells (arrowheads) detected by both 5'*Trpml3* (B) and 3'*Trpml3* (C) probes. **(F-K):** Immunohistochemistry using nonfluorescent detection (ABC+DAB) shows NT and CT1 antibodies specifically label scattered cells (arrowheads) in sections of adult (P48) *Trpml3*$^{+/+}$ (F-I) but not *Trpml3*$^{-/-}$ lungs (J, K). **(L, M):** Co-immunohistochemistry on sections of adult (P48) *Trpml3*$^{+/+}$ lungs indicates that NT colabels F4/80 positive macrophages. Scale bars indicate 50 μm in D, E, F, G, J, K and 10 μm in H, I, L, M.

raised against the cell surface macrophage marker F4/80, we found that NT-immunoreactive cells co-expressed F4/80 (Fig 2L and 2M).

Together, these data confirm that TRPML3 is expressed in alveolar macrophages in the lung, as previously demonstrated using the *Trpml3*$^{IRES-Cre/eR26-\tau GFP}$ reporter mouse model [32], and that its expression correlates with the onset of lung function after birth, when mature alveolar macrophages populate the lung [38].

## Olfactory bulb and olfactory receptor neurons show high TRPML3 expression

*In situ* hibridization and immunohistochemistry previuosly revealed prominent expression of Trpml3 mRNA and protein in olfactory receptor neurons [30]. These neurons, whose somas reside in nasal neuroepithelia, send axonal processes towards the brain's olfactory bulb. Hence we examined the olfactory bulb of *Trpml3*$^{IRES-Cre/eR26-\tau GFP}$ reporter mice, in which GFP is fused to the microtubule-associated protein Tau (τGFP) thereby assuring fluorescent labeling of neuronal axons [34, 39]. Hence, we prepared coronal sections from the whole brain, detecting dense τGFP expressing fibres in the outer layer of the olfactory bulb (= glomerular layer) (Fig 3A and 3B). The glomerular layer contains spherical structures called olfactory glomeruli, where synapses are formed between terminals of the olfactory receptor neurons (ORNs), which are located in the nasal epithelium, and dendrites of mitral cells having their soma in the mitral cell layer (MCL). τGFP expression is clearly visible in glomeruli (Fig 3C and 3D), indicating TRPML3 expression in these structures. Since TRPML3 seemed to be absent in mitral cells of the MCL, we assumed that the presence of τGFP in the glomeruli must stem from protruding axons and terminals of the ORNs, located in the nasal epithelium. Hence, we analysed expression in the nose (Fig 3E–3H). An overview image already pointed to TRPML3 expression in and underneath the olfactory epithelium (OE) (Fig 3E). Zoomed images of the nasal septum and OE confirmed the presence of τGFP, especially high signal could be seen in areas underneath the OE, in the lamina propria, where plenty of nerve fibres of the ORNs are bundled before projecting to the olfactory glomeruli (Fig 3F–3H). The coincidence of *Trpml3* expression pattern previously reported by *in situ* hybridization and immunohistochemistry with that revealed with the *Trpml3*$^{IRES-Cre/eR26-\tau GFP}$ mouse supports the validity of using this reporter line to deterime the expression of TRPML3.

## Skin melanocytes express TRPML3

Two prior reports indicated expression of *Trpml3* in melanocytes. First, Varitint-waddler (Va and VaJ) mice exhibited strong pigmentation defects, resulting putatively from the death of melanocytes [15]. Second, TRPML3 immunoreactivity was shown in cells of the mouse hair follicle, which immunoreacted to HMB-45, a known melanoma/melanosome marker [31].

We prepared cryosections visualizing the back skin of an adult *Trpml3*$^{IRES-Cre/eR26-\tau GFP}$ reporter mouse, which revealed high τGFP signal in hair follicles (Fig 4A). Moreover, to

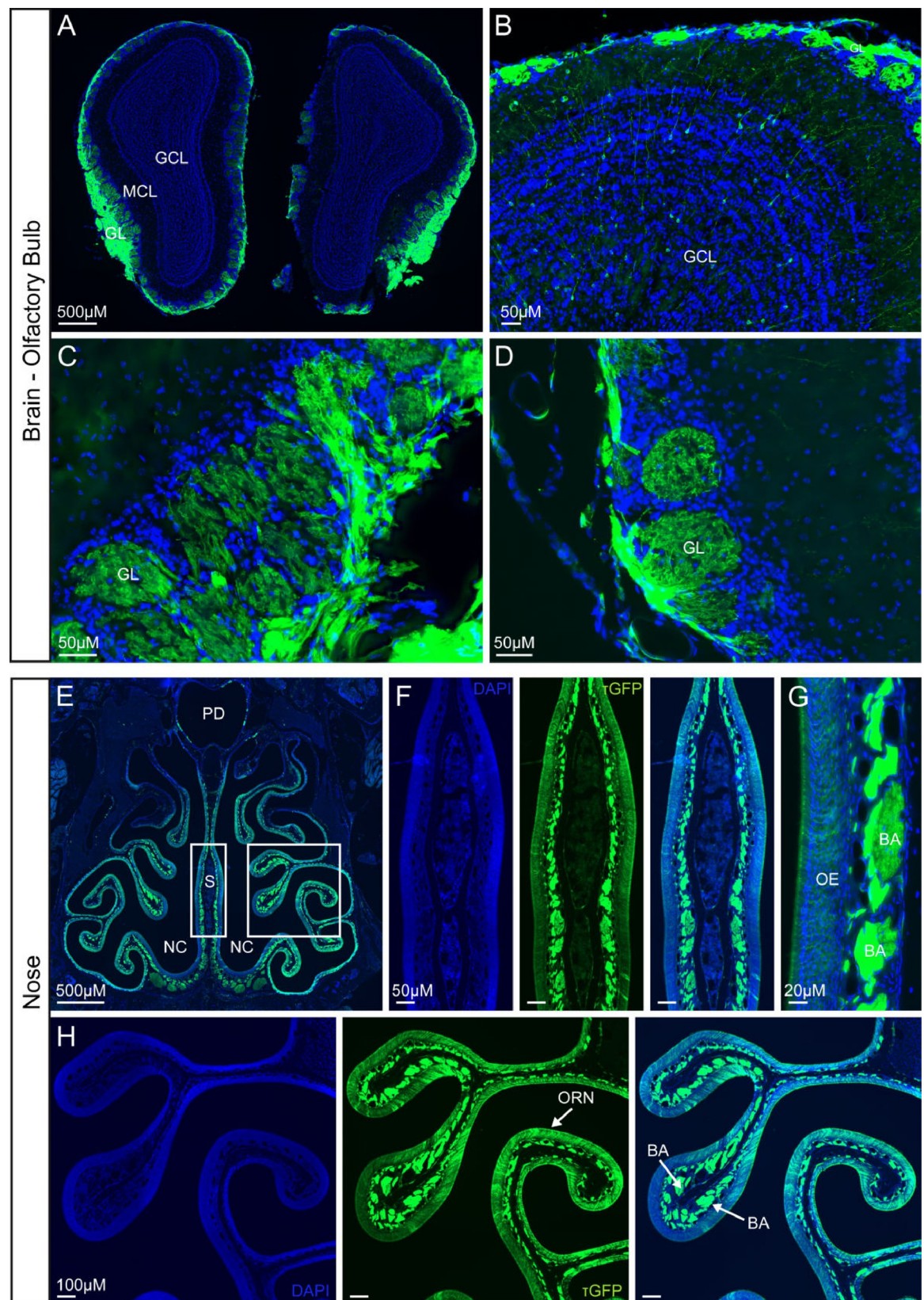

**Fig 3. TRPML3 distribution in olfactory bulb and nose. (A, B)** Overview images of coronal sections of the olfactory bulb. GL, glomerular layer. MCL, mitral cell layer. GCL, granule cell layer. τGFP expression is mainly detected in the GL. **(C, D)** Magnified images of the glomerular layer (GL) showing prominent τGFP expression in the glomeruli. **(E)** Coronal section of the nose. NC, nasal cavity. PD, pharyngeal duct (nasopharynx). S, nasal septum. **(F-H)** Zoomed images of the nasal septum (S) and olfactory epithelium (OE). τGPF signal is found in olfactory receptor neurons (ORNs) of the olfactory epithelium (OE) and in the bundles of axons (BA) that are projecting to the glomeruli of the olfactory bulb.

correlate reporter gene expression with *Trpml3* expression, we analyzed *Trpml3* mRNA levels by RT-qPCR and observed that *Trpml3* levels are in line with the hair follicle growth cycle (Fig 4B). *Trpml3* levels were highest during periods of hair follicle growth: anagen I (P9) and anagen II (P27) and low, though still reliably detected, during the period of hair follicle destruction and dormancy: catagen or telogen stages (P20). When we analyzed sections of back skin synchronized in anagen I (Fig 4C, 4D and 4G) and anagen II (Fig 4E) using both non-overlapping antisense ISH probes to *Trpml3* mRNA, we detected signal in a population of BrdU-negative, non-dividing cells located just above the dermal papilla (Fig 4F and 4G). When we performed immunohistochemistry using NT antisera, we observed immunoreactivity in cells that also express the melanocyte marker tyrosinase (Fig 4H, 4J, 4L–4O). This TRPML3-NT immunoreactivity is specific to TRPML3, as it was not detected in *Trpml3*$^{-/-}$ hair follicles of identical age (Fig 4I and 4K). TRPML3 immunoreactivity is seen in tyrosinase-positive melanocytes in hair follicles (Fig 4H–4M) and tyrosinase-positive melanocytes in glabrous (non hairy) skin (Fig 4N and 4O). Tyrosinase immunoreactivity does not appear grossly altered in the *Trpml3*$^{-/-}$ hair follicle (Fig 4J and 4K). From these data we conclude that TRPML3 is expressed in melanocytes of both glabrous skin and of hair follicles, showing highest expression in hair follicles during growth cycle phases anagen I and II, when melanogenesis takes place [40].

## Principle cells of the collecting duct in neonatal and adult kidney express TRPML3

Our initial RT-qPCR analysis showed that adult kidney expressed the highest levels of *Trpml3* mRNA of all tissues analyzed, except for RPE. Accordingly, we found high τGFP signal in kidney sections prepared from adult *Trpml3*$^{\text{IRES-Cre/eR26-τGFP}}$ reporter mice (Fig 5A–5C). The presence of τGFP was apparently rather high in areas, where the collecting ducts are found (Fig 5A and 5B). Some scattered τGFP signal was also detected in the renal cortex (Fig 5A and 5C).

We also used neonatal (P2) kidney sections and both non-overlapping antisense ISH probes to *Trpml3* mRNA (Fig 5D and S1 File). Our results clearly indicated *Trpml3* expression in either a subpopulation of kidney tubules or particular region(s) of the nephron unit. To determine where TRPML3 is expressed in the nephron, we performed double immunohistochemistry on sections of postnatal (P7 and P13) *Trpml3*$^{+/+}$ and *Trpml3*$^{-/-}$ kidney using our NT antisera together with the markers aquaporin 1 (which labels descending thin limb), and aquaporin 2 (which labels principle cells of the cortical, outer medullary and inner medullary collecting duct) [41, 42]. "Our NT antisera immunoreacted with numerous tubes in the *Trpml3*$^{+/+}$ kidney, but only a subset of these immunoreactivities was removed in the *Trpml3*$^{-/-}$ kidney. In prior analyses, we have seen certain immunoreactivities in other tissues that were not removed by the *Trpml3*$^{-/-}$ (30) and concluded they were non-specific (detecting a protein other than TRPML3). Therefore, we focused our attention on tubes whose NT immunoreactivities were removed in *Trpml3*$^{-/-}$ tissue, which are specific to TRPML3 (Fig 5G and 5I)". When we analyzed >110 NT-positive tubes that fit this criteria from three independent sections of each co-immunohistochemistry experiment, we found that TRPML3 is almost completely excluded from aquaporin 1 expressing tubes (1.77% of NT-positive tubes are also

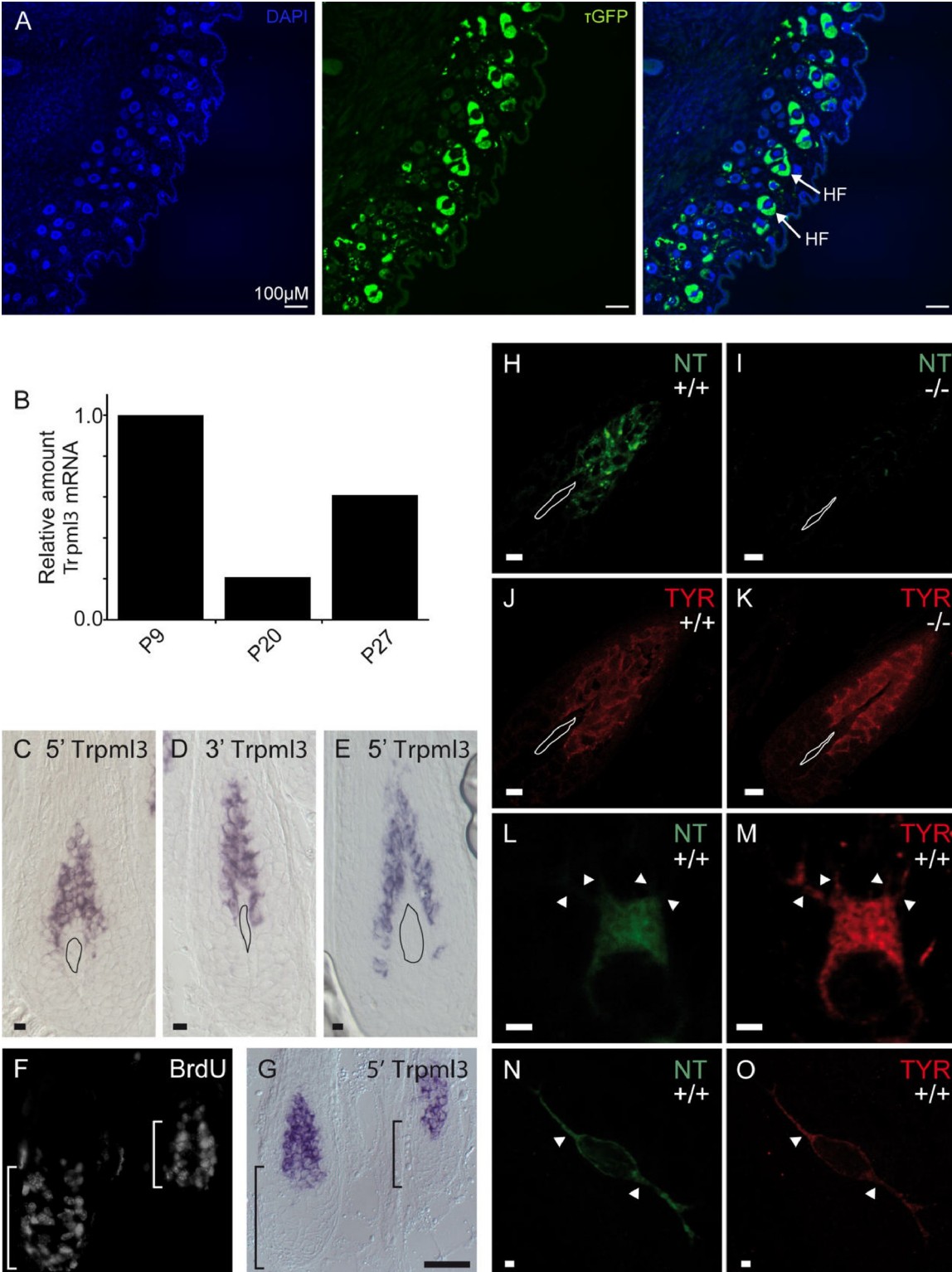

**Fig 4. Skin melanocytes express TRPML3. (A)** Images showing sections of the back skin. τGFP expression is visible in hair follicles (HF). **(B)** *Trpml3* levels in back skin correlate with initial synchrony of the hair follicle growth cycle. Follicle growth phases (anagen I, P9; anagen II, P27) show highest *Trpml levels* and the intervening follicle destruction phase (catagen/telogen, P20) shows the lowest *Trpml3* level. *Trpml3* levels detected and normalized similar to Fig 1 and displayed relative to normalized *Trpml3* levels in P9 back skin. **(C-E)** In situ hybridization on sections of CD1 (white) back skin using 5' (C,E) and 3' (D) probes for showing the presence of *Trpml3* in

anagen I (P10; C, D) and anagen II (P27; E) hair follicles. **(F, G)** BrdU immunohistochemistry performed with in situ hybridization shows *Trpml3* expressed in non dividing cells of anagen 1 (P6) hair follicles. **(H-O)** Immunohistochemistry on sections of P4-P7 (anagen I) *Trpml3*$^{+/+}$ (H, L, N) and *Trpml3*$^{-/-}$ (I) skin using an antibody raised against the amino-terminus of TRPML3 (NT) labels tyrosinase positive melanocytes in *Trpml3*$^{+/+}$ hair follicles (H, J, L, M) and glabrous skin (N, O) but not in *Trpml3*$^{-/-}$ follicles (I, K). H-K represent images from pigmented mice. Melanin was bleached to reveal immunoreactivity. L-O represent high magnification images from CD1 (white) mice of single melanocytes with dendritic processes indicated (arrowheads). Irregularly shaped objects on C-E and H-K indicate the dermal papilla. Scale bars indicate 10 μm in C-E and H-K; 50 μm in F, G; 2 μm in L-O.

Aquaporin 1-positive; Table 1, Fig 5E and 5F) and expressed in the same tubes and cells that express aquaporin 2 (99.1% of NT-positive tubes are also aquaporin 2-positive; Table 1, Fig 5G and 5H). We also examined the subcellular localization of TRPML3 in the aquaporin 2-positive principle cells of the collecting duct (Fig 5K–5P), where aquaporin 2 is known to reside in

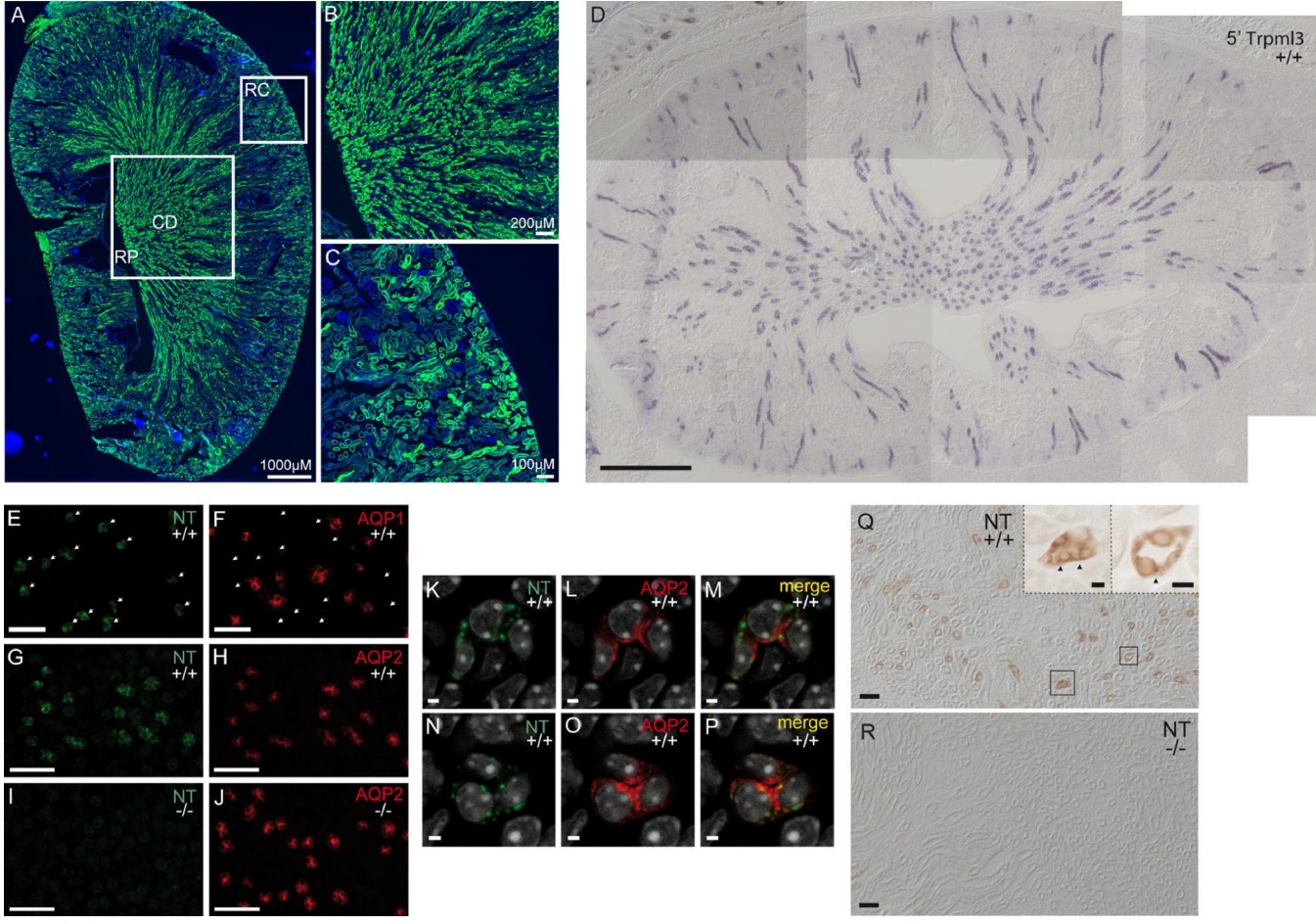

**Fig 5. Principle cells of the collecting duct in neonatal and adult kidney express TRPML3. (A)** Overview image of a longitudinal section of an adult kidney. RP, renal pelvis. RC, renal cortex. CD, collecting ducts. **(B)** Magnified image of the renal medulla with collecting ducts, which open into the renal pelvis. Cells of the collecting ducts seem to express τGFP. **(C)** Zoomed image of the renal cortex revealing renal tubules, of which some contain τGFP+ cells. **(D)** Assembled montage showing in situ hybridization on section of neonatal *Trpml3*$^{+/+}$ kidney using 5'*Trpml3* probe showing *Trpml3* expression in select tubes. **(E, F)** Immunohistochemistry on adjacent *Trpml3*$^{+/+}$ neonatal kidney sections using NT (E) and aquaporin 1 (F) antibodies shows that NT does not label the same tubes as aquaporin 1. **(G-J)** Co-immunohistochemistry shows that TRPML3 amino-terminal antibody NT colabels aquaporin 2 positive tubes in *Trpml3*$^{+/+}$ (G, H) but not *Trpml3*$^{-/-}$ (I, J) neonatal kidney. **(K-P)** Magnified, high resolution optical sections taken perpendicular to a single tube within the *Trpml3*$^{+/+}$ section photographed for G and H do not show a high degree of colocalization between TRPML3-positive and aquaporin 2-positive vesicles. **(Q, R)** Immunohistochemistry using nonfluorescent detection (ABC+DAB) on adult kidney sections reveal that NT also labels tubes in *Trpml3*$^{+/+}$ (N, insets) but not *Trpml3*$^{-/-}$ (O) adult kidney. Insets in N show high resolution of select tube cross sections (boxes) showing large vesicles (arrowheads). Ages of tissue used are: P2 (D), P13 (G-P), P7 (E, F), and P48 (Q, R). Scale bars indicate 500 μm (D), 50 μm (E-J, Q, R), 10 μm (insets on Q) and 2 μm (K-P).

intracellular vesicles until the vesicles are stimulated to fuse to the apical/luminal side of the principle cells, depositing aquaporin 2 in the plasma membrane. When we examined high resolution images of collecting duct cross sections, we observed NT immunoreactivity in vesicles in aquaporin 2-positive principle cells, however, each vesicle population appears to have a different subcellular localization. TRPML3 is expressed by principle cells of the collecting duct, but in vesicles other than the recycling endosomes that contain aquaporin 2.

We also performed immunohistochemistry on sections of adult (P48) kidney to see if TRPML3-specific immunoreactivity is maintained into adulthood. To accomplish this, we switched to a non-fluorescent detection method (ABC+DAB) so we could determine unambiguously if NT immunoreactivities we detected in adult *Trpml3*[+/+] kidney were removed in the *Trpml3*[-/-] kidney. Our analysis (Fig 5Q and 5R) shows that TRPML3 expression is maintained in large vesicles of principle cells of the collecting duct.

Thus far in all examined tissues, in which we found TRPML3-expressing cells by in situ hybridization and immunohistochemistry, the *Trpml3*[IRES-Cre/eR26-τGFP] reporter labels the same cell types: alveolar macrophages of lung, olfactory receptor neurons, melanocytes of skin, and principle cells of the collecting duct of kidney. In none of the tissues examied did we identify a cell type expressing *Trpml3* mRNA and protein that did not label with the τGFP reporter. Hence, we proceeded to using these mice to search for additional TRPML3-expressing cells in other tissues not previously examined for expression of endogenous TRPML3.

## TRPML3 distribution in thymus

*Trpml3* mRNA was also found in the thymus by RT-qPCR (Fig 1A). Additionally by using tissue sections form adult *Trpml3*[IRES-Cre/eR26-τGFP] reporter mice, we detected a set of τGFP signals throughout the whole thymus (Fig 6A). Co-stainings using antibodies to stain various immune cells were used to reveal the exact cell type expressing τGFP. A monoclonal CD8 alpha antibody identified cytotoxic/suppressor T-cells exclusively in the cortex of the thymus, which did not co-localize with τGFP (Fig 6B–6F). Similary, CD45R positive B-cells did not show co-localization with τGFP (Fig 6G–6J). We could only identifiy some single, F4/80 positive macrophages also expressing τGFP (Fig 6K–6N), but not all macrophages showed co-localization (Fig 6N), indicative of TRPML3 being present only in a subset of macrophages. Overall, we could exclude TRPML3 presence in cytotoxic T-cells and B-cells in the thymus, but confirmed TRPML3 expression in a small subset of macrophages.

## TRPML3 distibution in thyroid gland and parathyroid gland

To extend our TRPML3 distibution analysis to a broader range of organs we sectioned tissue from the *Trpml3*[IRES-Cre/eR26-τGFP] reporter mice containing parts of the trachea, oesophagus, thyroid and parathyoid (Fig 7A). Trachea, oesophagus and thyroid clearly showed no or only

**Table 1. Percent AQP2+ and AQP1+ tubes as a function of NT+ tubes.**

| Total NT+ tubes | % AQP2+ | % AQP2- |
|---|---|---|
| (117) | 99.1 (116) | 0.862 (1) |
| **Total NT+ tubes** | **% AQP1+** | **% AQP1-** |
| (113) | 1.77 (2) | 98.2 (111) |

TRPML3-NT positive tubes counted for three independent image pairs per double IHC. Each NT positive tube was then examined for co-expression of Aquaporin2 (AQP2), and Aquaporin1 (AQP1). For NT/AQP1 combinations, adjacent sections were stained and the tube cross section pattern aligned using Digital image correlation (DIC).

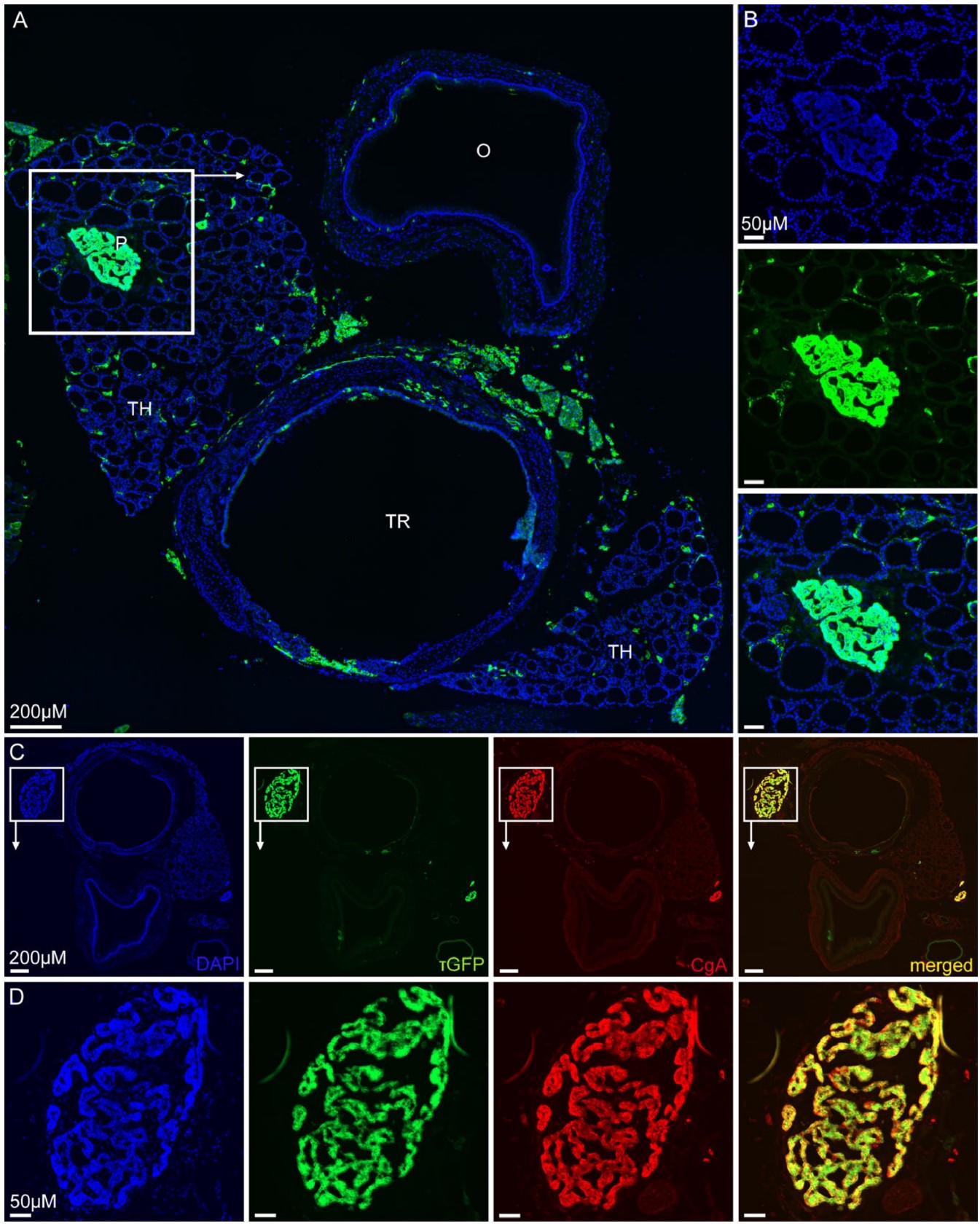

**Fig 6. TRPML3 distribution in thymus.** (A) Shown is an overview image of one lobe of the thymus. (B) Image of CD8 alpha stained whole thymus tissue, revealing regions of cytotoxic T-cells mainly in outer regions within the thymus. C, cortex. M, medulla. (C-F) Zoomed image of the region highlighted in C. (G-J) Thymus tissue stained with CD45R, revealing B-cells not colocalizing with τGFP. (K-N) Immunostaining using an antibody against F4/80 shows few macrophages expressing τGFP (arrowheads).

few τGFP+ cells. Areas of fluorescence emission between trachea and oesophagus is unspecific and originates from fatty-tissue-derived autofluorescence. Thyroid tissue mainly consists of spherical follicles, surrounded by a single layer of follicular cells, and some calcitonin-secreting parafollicular cells (= C cells), located in regions between the follicles. These circling follicular cells are easy to recognise throughout the thyroid tissue, but don't express TRPML3, since τGFP signal is absent in these cells (Fig 7A and 7B). Instead some scattered τGFP+ cells were detected between the follicles, which may represent parafollicular cells. Very dominant expression of τGFP was seen in the parathyroid gland (Fig 7A and 7B). We could identify the parathyroid gland within the thyroid tissue because of its different histology: densley packed cells in parathyroid compared to follicular structures of the thyroid (see Fig 7B; DAPI image). The parathyroid gland comprises two unique cell types: parathyroid hormone (PTH) secreting chief cells and the less abundant oxyphil cells. Oxyphil cells are found in humans but have not been described to occur in domestic animals [43, 44], e.g. being absent in dog, cat and rat [45] and were also not mentioned in studies on the mouse parathyriod [46, 47]. Hence, the high τGFP signal in the parathyroid is likely due to chief cells expressing TRPML3. We next performed double-immunofluorescence stainings of parathyroid sections using an antisera against Chromogranin A (CgA) (Fig 7C and 7D). CgA is a sensitive but unspecific marker for neuroendocrine cells and plays an important role in the biogenesis of secretory granules [48]. Antibodies directed against CgA to stain parathyroid tissue have already been used before [49, 50]. In our stainings the CgA antibody uniquely stained the parathyroid tissue, which nicely co-localizes with the τGFP expression (Fig 7C and 7D). To conclude, TRPML3 in the parathyroid is present in chief cells, whereas it is mostly absent from the thyroid.

## TRPML3 distribution in salivary, adrenal, and pituitary glands

Next, we analyzed further glands to find out if TRPML3 is present in other (neuro)endocrine or secretory cells. We again used tissue sections prepared from adult $Trpml3^{\text{IRES-Cre/eR26-τGFP}}$ reporter mice. An overview image of the salivary gland showed scattered τGFP+ cells throughout the different lobules (Fig 8A). Since the cervical lymph nodes are located close to the salivary glands, we were also able to visualize high τGFP signal in the lymph nodes (Fig 8B). Higher magnification images of the salivary gland tissue revealed the typical structures of secretory acini, made up by several secretory cells, releasing serous or mucous into the encircled duct. Some of these acini seem to express τGFP (Fig 8C). A co-staining using CgA antibody on salivary gland tissue apparently revealed some secretory acini with higher CgA expression than others. Interestingly, the areas with more intensively red CgA stained structures are overlapping with those that are postitive for τGFP (Fig 8D–8F). This suggests TRPML3 being present in cells or structures, that have some secretory function, similar to cells of the parathyroid.

The adrenal glands are located on top of the kidneys (Fig 8G). They can be histologically divided into the adrenal cortex, made up of cells secreting steroid hormones (aldosterone, cortisol, androgens), and the adrenal medulla, which contains neuroendocrine cells (= chromaffin cells), secreting catecholamines (adrenaline, noradrenaline). As expected and also reported previously [51] chromaffin cells of the adrenal medulla stained positive for CgA, whereas adjacent cells of the adrenal cortex were negative (Fig 8H–8K). τGFP expression, on the other

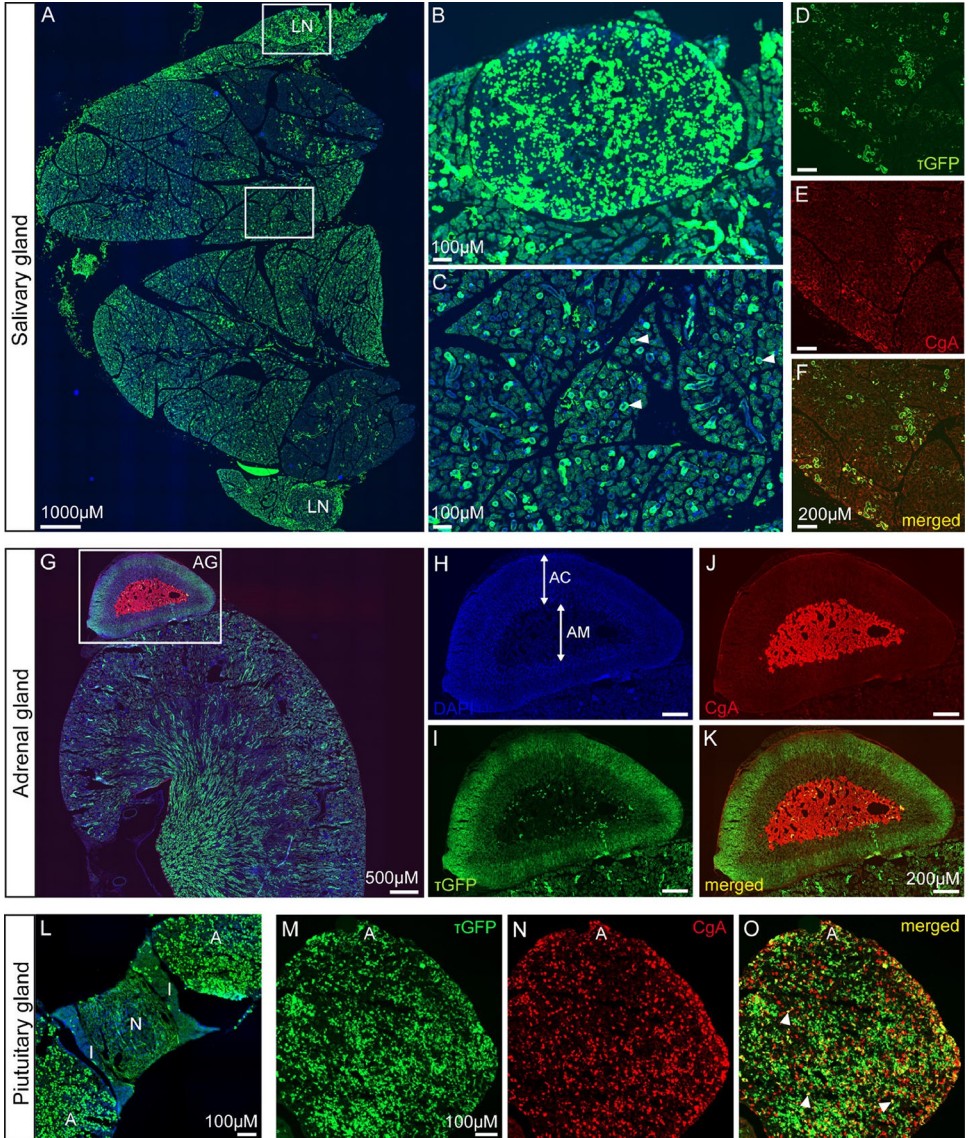

**Fig 7. TRPML3 distribution in thyroid and parathyroid. (A)** Whole tissue section showing the oesophagus (O), trachea (TR) and adjacant thyroid (TH), including parathyroid (P). **(B)** Zoomed image of the highlighted region of (A) shows the parathyroid gland with a noticable high τGFP expression. **(C, D)** Immunostaining using an antibody against chromogranin A (CgA) reveals neuroendocrine cells of the mouse parathyroid in red. τGFP+ cells perfectly overlap with CgA+ cells.

hand, was mainly found in outer zones of the adrenal cortex, while it was only negligibly present in the adrenal medulla (Fig 8H–8K).

Finally, we analysed the pituitary gland. The overview picture allows distinction between the different lobes: the adenohypophysis, the intermediate lobe and the neurohypophysis (Fig 8L). τGFP+ cells are mainly seen in the adenohypophysis (Fig 8L), which contains a variety of different cell types that synthesize and secrete hormones, e.g. somatotropes secreting human growth hormone, corticotropes secreting adrenocorticotropin, thyrotropes secreting thyroid stimulating hormone, gonadotropes secreting gonadotropic hormones (LH and FSH) and lactotropes secreting prolactin. An immunostaining using CgA antibody positively stained a large

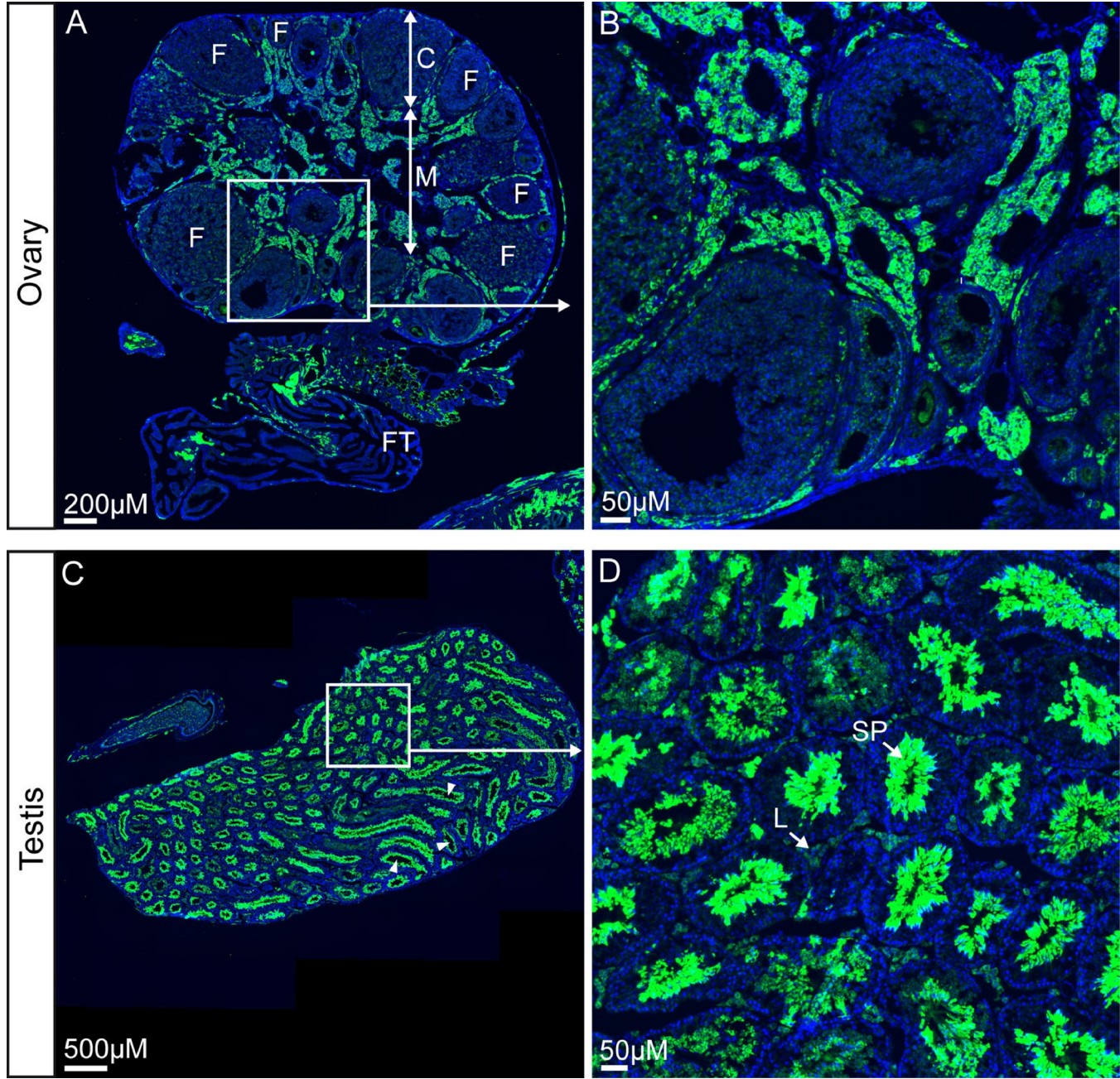

**Fig 8. TRPML3 distribution in salivary gland, adrenal gland, and pituitary gland. (A)** Overview image of the salivary gland with its lobules. Two submandibular lymph nodes (LN) are located at the sites. **(B)** Zoomed image of the upper lmph node (LN). **(C)** Zoomed image of the salivary gland tissue reveals the typical structure of secretory acini (arrowheads), consisting of several secretory cells. Some of these acini show τGFP expression. **(D-F)** Immunostainings show CgA expression in secretory cells. Some of the secretory cells building up one acinus reveal higher CgA expression than others. This higher CgA expression is colocalizing with τGFP (merged/yellow). **(G)** Immunostaining of the kidney with the adrenal gland (AG) sitting on top using CgA to label neuroendocrine cells. **(H-K)** Magnified images of the adrenal gland from (G). DAPI staining reveals the adrenal medulla (AM) and the adrenal cortex (AC). CgA antibody stains the catecholamine secreting chromaffin cells in the adrenal medulla. τGFP+ cells are nearly absent from the adrenal medulla, but seem to be more prominent in outer regions of the adrenal cortex. **(L)** Overview image of the pituitary gland. A, adenohypophyis. N, neurohypophysis. I, intermediate lobe. τGFP expression was mainly found throughout the adenohypophysis. **(M-O)** Zoomed image and immunostaining of the adenohypophysis using an antibody against CgA reveals several endocrine cells (CgA+ cells) overlapping with τGFP (arrowheads).

quantity of endocrine cells in the adenohypophysis, of which however only few cells co-localized with τGFP (Fig 8M–8O). Only low τGFP signal was detected in the neurohypophysis (Fig 8L). It is likely that the signal originates from neuronal fibers projecting from the median eminence to the pituitary. Further studies need to elaborate on the hypothalamic neuron population from which these fibres emanate.

## To summarize, TRPML3 is present in the salivary, adrenal, and pituitary glands, but the exact cell types expressing TRPML3 remain to be further defined

TRPML3 distribution in ovary and testes

To complete the study we did an analysis of the reproductive organs, i.e. ovary and testes. Longitudinal sections of the ovaries of adult $Trpml3$^IRES-Cre/eR26-τGFP reporter mice, including fallopian tubes pointed to τGFP expression in the medulla of the ovary (Fig 9A and 9B), which mainly consists of lymphatics, nerves and numerous blood vessels. No τGFP fluorescence was seen in maturing follicles that are located in the cortex of the ovary (Fig 9A and 9B).

An overview picture of the testis shows many seminiferous tubules (Fig 9C), which is the location of sperm cell (= spermatozoa) production through differentiation of spermatogenic cells. The higher magnification image shows that these sperm cells in the middle of the seminiferous tubules are positive for τGFP (Fig 9D). Testosterone-producing Leydig cells (L), which are located adjacent to the seminiferous tubles in the interstitium did not show significant τGFP signal.

## Discussion

In this study, we performed a detailed expression analysis of various mouse tissues for the endolysosomal TRP channel TRPML3, using several methodological approaches. We confirmed the presence of $Trpml3$ in VNO, RPE, thymus, spleen, back skin, kidney and lung by RT-qPCR as described previously [17, 24, 28, 30], but now complemented this with data gained from in situ hybridization (ISH) and immunohistochemistry (IHC) using $Trpml3$^+/+ and $Trpml3$^-/- control mice and immunofluorescence using tissue sections of a reporter mouse model ($Trpml3$^IRES-Cre/eR26-τGFP) [32]. A similar mouse model for TRPM5 and TRPV6 has been developed before and was successfully used to describe the TRPM5 and TRPV6 expression pattern throughout the murine body [33, 52]. This experimental approach of genetic cell labeling in mice offers the advantage of an antibody-independent visualization of the protein of interest without the problems associated with antibody generation and specificity. For TRP channels, good antibodies are often simply not available. Here, we used both, antibodies against TRPML3 with knockout controls and the $Trpml3$^IRES-Cre/eR26-τGFP reporter mouse model to identify TRPML3+ cells. The results that we obtained from IHC and ISH experiments generally correlated well with the images gathered from the reporter mouse sections: this specifically applies to TRPML3 in skin melanocytes/hair follicles, in the collecting ducts of the kidney, in lung alveolar macrophages and in olfactory sensory neurons as also published previously [30]. With regard to alveolar macrophages, the TRPML3 channel activity could be recently confirmed by endolysosmal patch clamp experiments using τGFP+ versus (the rarely occuring) τGFP- alveolar macrophages and ML3-SA1, a selective activator of the TRPML3 channel [32]. Generally, the use of various approaches makes this expression study both more interesting and importantly more reliable. One important point to consider regarding the reporter mouse is, however, that statements about developmental timing of channel expression cannot be made, i.e. if a cell at a single and early point in its development expresses the channel, the strong stop signal within the Rosa26 locus is permanently removed, and τGFP

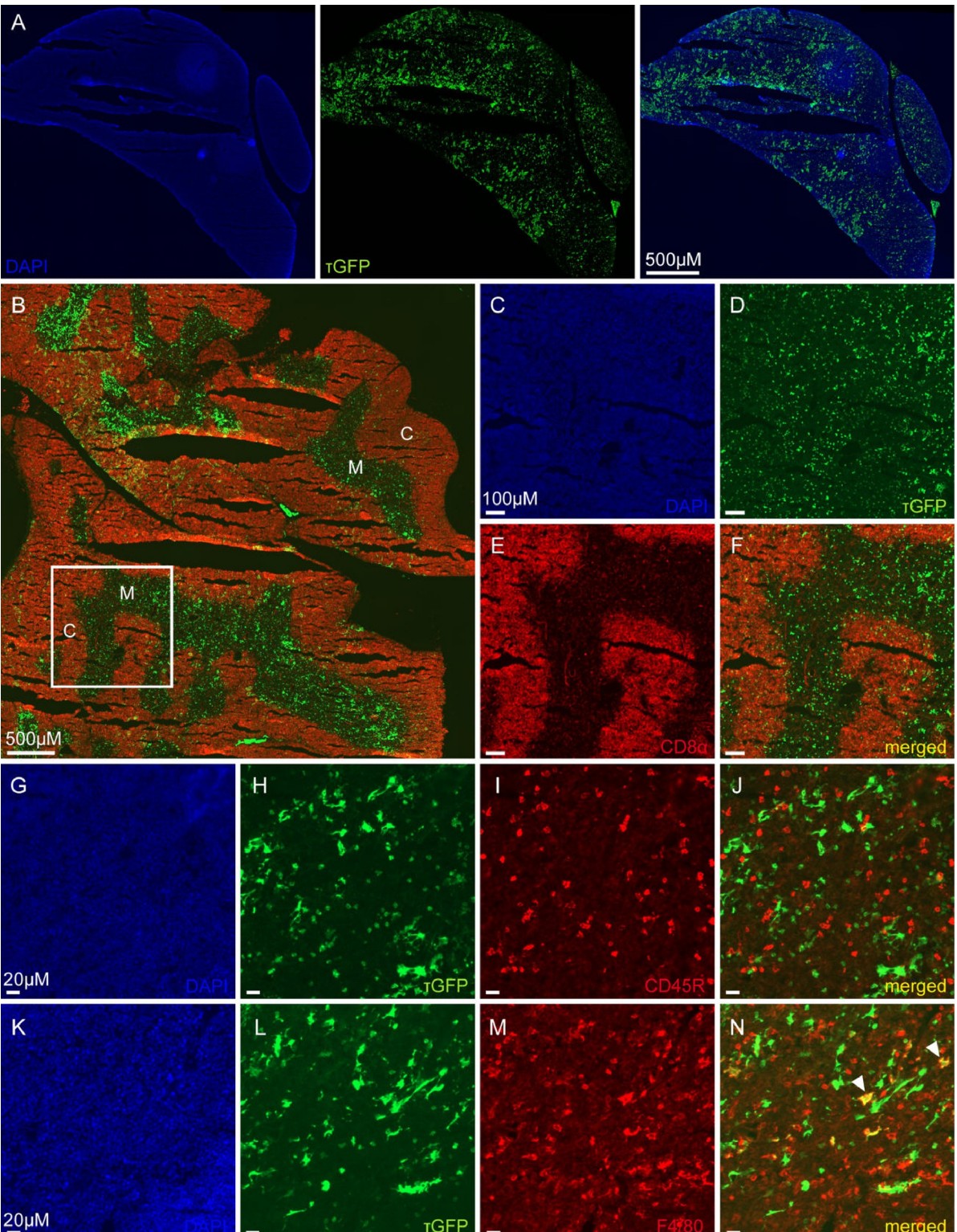

**Fig 9. TRPML3 distribution in ovary and testis. (A)** Shown is a longitudinal section of the ovary and fallopian tube (FT). The cortex (C) contains numerous follicles (F) at various stages of maturation. τGFP fluorescence is mainly found in the medulla (M), which is consisting of lymphatics, nerves and numerous blood vessels. Adjacent to the ovary: the fallopian tube (FT). **(B)** Zoomed image of highlighted region in (A) shows parts of the medulla and some developing follicles of the ovary. **(C)** Overview image showing the testes with the seminiferous tubules (arrowheads). **(D)** Zoomed image of indicated area of (C) reveals spermatozoa (SP) in the middle of the seminiferous tubule being positive for τGFP. Leydig cells (L) are visible as interstitial cells adjacent to the seminiferous tubule.

expression will continue through to mark the expressing cells from this time point. That is because of the purpose of the Cre-reporter to label TRPML3+ cells by faithfully expressing τGFP in the cells and to open the availability to further mainpulate these cells with different ROSA reporter mouse strains. Anyways, to differentiate between expression in either early or later stages of development we therefore included both adult and prenatal or neonatal tissues for some of our IHC analyses.

In summary, with this expression analysis we have shown TRPML3 expression in various specific cell types such as lung alveolar macrophages, olfactory sensory neurons, skin melanocytes, and principle cells of the collecting duct of the kidney. Furthermore, we analyzed several gland tissues (thyroid/parathyroid gland, salivary gland, adrenal gland and pituitary gland), whose TRPML3 expression has not been demonstrated before. TRPML3 was found in the parathyroid gland, assumably in chief cells, but not in the thyroid. Of note, a similar expression scenario for TRPML3 in thyroid/parathyroid is found in humans when using transcriptome or proteome databases such as UniGene EST in Grimm et al. [24] or the Human Protein Atlas [53] (http://www.proteinatlas.org). Chief cells are responsible for the secretion of parathyroid hormone (PTH) and thereby regulate and maintain normal blood calcium levels. Further, we found TRPML3 in several other hormonal glands such as the cortex of the adrenal gland, the adenohypophysis of the pituitary gland and the testes, presumably in spermatozoa. The presence of TRPML3 is also again overlapping with a database screen using the Human Protein Atlas [53] (http://www.proteinatlas.org) that shows high *Trpml3* mRNA expression especially in adrenal gland and pituitary gland, as well as high TRPML3 expression in testes. Other mouse gene expression databases detect *Trpml3* expression in the brain, e.g. the Allen Brain Atlas (https://portal.brain-map.org/), or Mouse Brain Atlas from the Linnarsson Lab (http://mousebrain.org/). Where exactly TRPML3 is expressed in the brain and glands, what its function may be and whether lack of or mutations in the *Trpml3* gene lead to dysfunctions in some biological processes associated with these glands, needs to be determined.

Our study does not claim to be complete, but it provides a first, broad overview of TRPML3 expression throughout the mouse body. Furthermore, the reporter mouse model used here will be made available for the research community to broaden our understanding of the physiology and pathophysiology of this channel in both mouse and man.

## Supporting information

**S1 File.**
(PDF)

## Acknowledgments

We would like to thank Drs. Markus Delling and David Clapham for the kind gift of TRPML3-CT1 antisera.

## Author Contributions

**Conceptualization:** Jaime García-Añoveros, Christian Grimm.

**Funding acquisition:** Natalie N. Remis, Ulrich Boehm, Thomas Gudermann, Martin Biel, Jaime García-Añoveros, Christian Grimm.

**Investigation:** Barbara Spix, Andrew J. Castiglioni, Natalie N. Remis, Emma N. Flores, Philipp Wartenberg.

**Methodology:** Andrew J. Castiglioni, Philipp Wartenberg, Amanda Wyatt, Ulrich Boehm.

**Resources:** Ulrich Boehm, Martin Biel, Jaime García-Añoveros, Christian Grimm.

**Supervision:** Jaime García-Añoveros, Christian Grimm.

**Visualization:** Barbara Spix, Andrew J. Castiglioni, Philipp Wartenberg.

**Writing – original draft:** Barbara Spix, Andrew J. Castiglioni, Jaime García-Añoveros, Christian Grimm.

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
