## [Decision Letter · Decision Letter 0]

19 Oct 2022

PONE-D-22-24077Whole-body analysis of TRPML3 expression using a GFP-reporter mouse model reveals widespread expression in secretory cells and endocrine glandsPLOS ONE

Dear Dr. Grimm,

Thank you for submitting your manuscript to PLOS ONE. After careful consideration, we feel that it has merit but does not fully meet PLOS ONE’s publication criteria as it currently stands. Therefore, we invite you to submit a revised version of the manuscript that addresses the concerns of the reviewers.

We look forward to receiving your revised manuscript.

Kind regards,

Alexander G. Obukhov, Ph.D.

Academic Editor

PLOS ONE

Journal Requirements:

"Financial Support: RO1 DK111032 and R01 DC015903 (to JGA), T32 NRSA NS041234 (to AJC and ENF), F31 NRSA DC010529 (to NNR), German Research Foundation (GRK2338 P08 to CG and MB, P09 to TG, SFB/TRR152 Z02 to UB, P04 to CG, P12 to MB, P15 and the German Center of Lung Research, DZL, to TG)."

"Financial Support: RO1 DK111032 and R01 DC015903 (to JGA), T32 NRSA NS041234 (to AJC and ENF), F31 NRSA DC010529 (to NNR), German Research Foundation (GRK2338 P08 to CG and MB, P09 to TG, SFB/TRR152 Z02 to UB, P04 to CG, P12 to MB, P15 and the German Center of Lung Research, DZL, to TG).

Reviewers' comments:

Reviewer's Responses to Questions

**Comments to the Author**

1. Is the manuscript technically sound, and do the data support the conclusions?

Reviewer #1: Yes

Reviewer #2: Yes

2. Has the statistical analysis been performed appropriately and rigorously? 

Reviewer #1: Yes

Reviewer #2: N/A

3. Have the authors made all data underlying the findings in their manuscript fully available?

Reviewer #1: Yes

Reviewer #2: Yes

4. Is the manuscript presented in an intelligible fashion and written in standard English?

Reviewer #1: Yes

Reviewer #2: Yes

5. Review Comments to the Author

Reviewer #1: This study systematically applied several methodological approaches to validate the expression patterns of the endolysosomal TRP channel, TRPML3, in various mouse tissues. The authors firstly confirmed the presence of TRPML3 in lung alveolar macrophages, olfactory sensory neurons, skin melanocytes, and principle cells of the collecting duct of the kidney. Secondly, the authors made new discovery on that TRPML3 is also expressed in thyroid/parathyroid, salivary, adrenal and pituitary gland. The confocal images shown in the study are clear and conclusive.

On the whole, this work provides the basic knowledge on the expression pattern of TRPML3 throughout the mouse body, which should be highly appreciated.

To further improve this manuscript, the following minor concerns should be addressed:

1. The statistical columns (Fig. 1A, 2C and 4B) lack of error bars.

2. In this study, two antisera raised against different regions of TRPML3 (NT and CT1) were used for detecting TRPML3 expression. However, the CT1 antiserum was adopted only once when examining adult lungs (Fig. 2), while NT antiserum appeared more frequent application. Is there any reason for the preference?

3. How to explain the observation that only a subset of NT antiserum immunoreactivities was removed from the Trpml3-/- kidney?

4. Trpml3 mRNA expression of B cells in thymus was analyzed. How about plasma cells?

Reviewer #2: The paper is methodologically sound and contains large amount of data that can be useful for researchers interested in various tissues. Below are listed some comments.

1) The official name of the gene is Mcoln3 (https://www.ncbi.nlm.nih.gov/gene/171166). Some researchers may prefer the name TRPML3 but more and more researchers will search for the official name. Therefore, add in parentheses the official name of the gene in the title of the paper. It is an easy way to increase the impact of the paper.

2) In addition to papers specifically studying TRPML3 (Mcoln3) there are also online databases showing pattern of expression of various genes. Two databases containing information about expression of Mcoln3 are:

http://mouse.brain-map.org/gene/show/82359

http://mousebrain.org/adolescent/genesearch.html

Please, include these databases in discussion (they should be cited together with papers that for the first time described these databases (https://doi.org/10.1038/nature05453 and https://doi.org/10.1016/j.cell.2018.06.021)

3) It is not clear how many mice were used to obtain the data. This information can be provided in figure descriptions (separately for each method).

6. PLOS authors have the option to publish the peer review history of their article (what does this mean?). If published, this will include your full peer review and any attached files.

Reviewer #1: **Yes: **Wuyang Wang

Reviewer #2: No

---

## [Author Response · Author response to Decision Letter 0]

14 Nov 2022

PONE-D-22-24077

Whole-body analysis of TRPML3 expression using a GFP-reporter mouse model reveals widespread expression in secretory cells and endocrine glands

PLOS ONE

Dear Dr. Grimm,

Thank you for submitting your manuscript to PLOS ONE. After careful consideration, we feel that it has merit but does not fully meet PLOS ONE’s publication criteria as it currently stands. Therefore, we invite you to submit a revised version of the manuscript that addresses the concerns of the reviewers.

• A rebuttal letter that responds to each point raised by the reviewers. You should upload this letter as a separate file labeled 'Response to Reviewers'.

We look forward to receiving your revised manuscript.

Kind regards,

Alexander G. Obukhov, Ph.D.

Academic Editor

PLOS ONE

Journal Requirements:

Authors: Checked

Authors: Done

"Financial Support: RO1 DK111032 and R01 DC015903 (to JGA), T32 NRSA NS041234 (to AJC and ENF), F31 NRSA DC010529 (to NNR), German Research Foundation (GRK2338 P08 to CG and MB, P09 to TG, SFB/TRR152 Z02 to UB, P04 to CG, P12 to MB, P15 and the German Center of Lung Research, DZL, to TG)."

"Financial Support: RO1 DK111032 and R01 DC015903 (to JGA), T32 NRSA NS041234 (to AJC and ENF), F31 NRSA DC010529 (to NNR), German Research Foundation (GRK2338 P08 to CG and MB, P09 to TG, SFB/TRR152 Z02 to UB, P04 to CG, P12 to MB, P15 and the German Center of Lung Research, DZL, to TG).

Authors: Please update the online form according to the indicated funders above. Thanks.

Authors: Removed

Authors: Done

Reviewers' comments:

Reviewer's Responses to Questions

Comments to the Author

1. Is the manuscript technically sound, and do the data support the conclusions?

Reviewer #1: Yes

Reviewer #2: Yes

2. Has the statistical analysis been performed appropriately and rigorously? 

Reviewer #1: Yes

Reviewer #2: N/A

3. Have the authors made all data underlying the findings in their manuscript fully available?

Reviewer #1: Yes

Reviewer #2: Yes

4. Is the manuscript presented in an intelligible fashion and written in standard English?

Reviewer #1: Yes

Reviewer #2: Yes

5. Review Comments to the Author

Reviewer #1: This study systematically applied several methodological approaches to validate the expression patterns of the endolysosomal TRP channel, TRPML3, in various mouse tissues. The authors firstly confirmed the presence of TRPML3 in lung alveolar macrophages, olfactory sensory neurons, skin melanocytes, and principle cells of the collecting duct of the kidney. Secondly, the authors made new discovery on that TRPML3 is also expressed in thyroid/parathyroid, salivary, adrenal and pituitary gland. The confocal images shown in the study are clear and conclusive.

On the whole, this work provides the basic knowledge on the expression pattern of TRPML3 throughout the mouse body, which should be highly appreciated.

Authors: We thank the reviewer for his positive comments. We have addressed his suggestions as outlined below.

To further improve this manuscript, the following minor concerns should be addressed:

1. The statistical columns (Fig. 1A, 2C and 4B) lack of error bars.

Authors: 1. This is because these RT-qPCRs were done each from a single source (organ for one animal). We did them in triplicate, as is customary for RT-qPCR in order to confirm the results are reproducible, but adding error bars would not make statistical sense. Accordingly, we did not make any statistical analyses from these graphs. We used them as a first pass indicator of which organs and at what stages may express Trpml3, and then proceed to confirm these results by in situ hybridization probes and immunohistochemistry in many samples of the relevant organ (in addition to the Trpml3IRES-Cre/eR26-τGFP reporter mice). We are attaching the excel spreadsheet with the data for these three RT-qPCR graphs. We could add error bars by separately calculating the data for each replicate, but these are not biological replicates and hence this would be IOO misleading.

2. In this study, two antisera raised against different regions of TRPML3 (NT and CT1) were used for detecting TRPML3 expression. However, the CT1 antiserum was adopted only once when examining adult lungs (Fig. 2), while NT antiserum appeared more frequent application. Is there any reason for the preference?

Authors: The only reason for using more often the NT antiserum is availability, as it is commercially available, whereas the CT1 antiserum is a gift from Markus Delling and David Clapham and, as such, we have limited amounts of it. However, we have used both antibodies in multiple tissues in addition to lung, such as cochlea (hair cells and principal cells of the stria vascularis), olfactory epithelium and vomeronasal organ chemosensory neurons (Castiglioni et al., 2011. J Comp Neurol).

3. How to explain the observation that only a subset of NT antiserum immunoreactivities was removed from the Trpml3-/- kidney?

Authors: The immunoreactivities that remain in the Trpml3 KOs are non-specific (cross reacting with proteins other than TRPML3). We have seen this in other organs, in which an antibody labels a cell type in the KO (hence in the absence of TRPML3) and that cell type does not express Trpml3 mRNA as determined by in situ hybridization.

We have clarified this by modifying the following statement: "Our NT antisera immunoreacted with numerous tubes in the Trpml3+/+ kidney, but only a subset of these immunoreactivities were removed in the Trpml3-/- kidney. In prior analyses, we have seen certain immunoreactivities in other tissues that were not removed by the Trpml3-/- (30) and therefore we focused our attention on tubes whose NT immunoreactivities were removed in Trpml3-/- tissue and specific to TRPML3 (Fig. 5G, I)". Instead, we now simply state that: "Our NT antisera immunoreacted with numerous tubes in the Trpml3+/+ kidney, but only a subset of these immunoreactivities was removed in the Trpml3-/- kidney. In prior analyses, we have seen certain immunoreactivities in other tissues that were not removed by the Trpml3-/- (30) and concluded they were non-specific (detecting a protein other than TRPML3). Therefore, we focused our attention on tubes whose NT immunoreactivities were removed in Trpml3-/- tissue, which are specific to TRPML3 (Fig. 5G, I)".

4. Trpml3 mRNA expression of B cells in thymus was analyzed. How about plasma cells?

Authors: We did not examine plasma cells, only solid tissues.

Reviewer #2: The paper is methodologically sound and contains large amount of data that can be useful for researchers interested in various tissues. Below are listed some comments.

1) The official name of the gene is Mcoln3 (https://www.ncbi.nlm.nih.gov/gene/171166). Some researchers may prefer the name TRPML3 but more and more researchers will search for the official name. Therefore, add in parentheses the official name of the gene in the title of the paper. It is an easy way to increase the impact of the paper.

Authors: We thank the reviewer for his comment and have added Mcoln3 as recommended.

2) In addition to papers specifically studying TRPML3 (Mcoln3) there are also online databases showing pattern of expression of various genes. Two databases containing information about expression of Mcoln3 are:

http://mouse.brain-map.org/gene/show/82359

http://mousebrain.org/adolescent/genesearch.html

Please, include these databases in discussion (they should be cited together with papers that for the first time described these databases (https://doi.org/10.1038/nature05453 and https://doi.org/10.1016/j.cell.2018.06.021)

Authors: This has been added accordingly.

3) It is not clear how many mice were used to obtain the data. This information can be provided in figure descriptions (separately for each method).

Authors: This info is now provided in the M&M section.

6. PLOS authors have the option to publish the peer review history of their article (what does this mean?). If published, this will include your full peer review and any attached files.

Do you want your identity to be public for this peer review? For information about this choice, including consent withdrawal, please see our Privacy Policy.

Reviewer #1: Yes: Wuyang Wang

Reviewer #2: No

---

## [Decision Letter · Decision Letter 1]

28 Nov 2022

Whole-body analysis of TRPML3 (MCOLN3) expression using a GFP-reporter mouse model reveals widespread expression in secretory cells and endocrine glands

PONE-D-22-24077R1

Dear Dr. Grimm,

We’re pleased to inform you that your manuscript has been judged scientifically suitable for publication and will be formally accepted for publication once it meets all outstanding technical requirements.

Kind regards,

Alexander G. Obukhov, Ph.D.

Academic Editor

PLOS ONE

Reviewers' comments:

Reviewer's Responses to Questions

**Comments to the Author**

1. If the authors have adequately addressed your comments raised in a previous round of review and you feel that this manuscript is now acceptable for publication, you may indicate that here to bypass the “Comments to the Author” section, enter your conflict of interest statement in the “Confidential to Editor” section, and submit your "Accept" recommendation.

Reviewer #1: All comments have been addressed

Reviewer #2: All comments have been addressed

2. Is the manuscript technically sound, and do the data support the conclusions?

Reviewer #1: Yes

Reviewer #2: Yes

3. Has the statistical analysis been performed appropriately and rigorously? 

Reviewer #1: Yes

Reviewer #2: N/A

4. Have the authors made all data underlying the findings in their manuscript fully available?

Reviewer #1: Yes

Reviewer #2: Yes

5. Is the manuscript presented in an intelligible fashion and written in standard English?

Reviewer #1: Yes

Reviewer #2: Yes

6. Review Comments to the Author

Reviewer #1: The authors have made a substantial revision and such effort is reflected in the manuscript.The authors' explanations satisfactorily addressed all my concerns.

Reviewer #2: I have no other comments for this manuscript.

7. PLOS authors have the option to publish the peer review history of their article (what does this mean?). If published, this will include your full peer review and any attached files.

Reviewer #1: **Yes: **Wuyang Wang

Reviewer #2: No

---

## [Editor Report · Acceptance letter]

5 Dec 2022

PONE-D-22-24077R1 

Whole-body analysis of TRPML3 (MCOLN3) expression using a GFP-reporter mouse model reveals widespread expression in secretory cells and endocrine glands 

Dear Dr. Grimm:

I'm pleased to inform you that your manuscript has been deemed suitable for publication in PLOS ONE. Congratulations! Your manuscript is now with our production department. 

Kind regards, 

on behalf of

Dr. Alexander G Obukhov 

Academic Editor

PLOS ONE